# Multiverse Predictions for Habitability: The Number of Stars and Their Properties

**McCullen Sandora** [1,2]

[1] Institute of Cosmology, Department of Physics and Astronomy, Tufts University, Medford, MA 02155, USA; mccullen.sandora@gmail.com

[2] Center for Particle Cosmology, Department of Physics and Astronomy, University of Pennsylvania, Philadelphia, PA 19104, USA

**Abstract:** In a multiverse setting, we expect to be situated in a universe that is exceptionally good at producing life. Though the conditions for what life needs to arise and thrive are currently unknown, many will be tested in the coming decades. Here we investigate several different habitability criteria, and their influence on multiverse expectations: Does complex life need photosynthesis? Is there a minimum timescale necessary for development? Can life arise on tidally locked planets? Are convective stars habitable? Variously adopting different stances on each of these criteria can alter whether our observed values of the fine structure constant, the electron to proton mass ratio, and the strength of gravity are typical to high significance. This serves as a way of generating predictions for the requirements of life that can be tested with future observations, any of which could falsify the multiverse scenario.

**Keywords:** multiverse; habitability; stars

## 1. Introduction

Science is beginning to embrace the idea of a multiverse—that is, that the laws of physics have the potential of being different elsewhere. In this framework, some of the parameters in our standard models of particle physics and cosmology vary from universe to universe, and are not capable of being explained mechanistically. This does not necessarily mean that there is no explanatory power of these theories; however, One of the requirements for a physical theory becomes that it allows sufficient complexity to give rise to what are termed observers, of which we as humans are presumably representative. These privileged, information processing-rich arrangements of matter are fragile, and so are extremely sensitive to the types of environments the underlying physics is capable of producing. A universe without our panoply of atomic states, for example, is expected to be devoid of sufficient complexity to give rise to these observers.

The mode of explanation we can hope for in such a scenario is to determine the chances of observing the laws of physics to be what they are, subject to the condition that we are typical observers. This style of reasoning usually goes by the name 'the principle of mediocrity' [1]. In order to employ it, it becomes necessary to try to quantify how many observers a universe with a given set of physical parameters is likely to host. Traditionally, cosmologists have shied away from the detailed criteria necessary for life and then intelligent life to emerge, largely because the specifics of the requirements are very uncertain at our current state of knowledge. Thus, attention has focused on the predictions for cosmological observables like the cosmological constant and density contrast [2–5], as opposed to quantities that may affect mesoscopic properties of observers.

In this context, a useful proxy for this complicated task has been to simply take the fraction of baryons that ended up in galaxies above a certain threshold mass, with the understanding that this

is necessary in order for heavy elements to be synthesized and recycled into another generation of stars and planets. However, this crude method, while useful for determining preferred values of cosmological parameters, has really only resulted in a rather limited number of mostly postdictions, such as the need to live in a universe which is big, old, empty, and cold.

While many anthropic boundaries regarding microscopic physical parameters such as the proton mass, electron mass, fine structure constant, and strength of gravity have been delineated [6–8] (for a recent review see [9]), comparatively little attention has been paid to these in the context of the principle of mediocrity. However, these largely determine many of the properties of our mesoscopic world, and so the details of the microphysical parameters will ultimately dictate quite strongly which universes will be capable of supporting observers. Placing this further level of realism on our estimations, however, requires a refined understanding of habitability. While still a major open issue, over the past few decades science has made amazing progress in understanding this question: We now have a much clearer view of the architecture of other planetary systems [10], we have discovered the ubiquity of preorganic chemical complexes throughout the galaxy [11], the outer reaches of our own solar system have revealed remarkable complexity [12,13], we understand the formation of planetary systems to unprecedented levels, and we now have atmospheric spectroscopy of nearly a dozen extrasolar planets (albeit mostly hot Jupiters) [14]. As amazing as this progress has been, the coming decades are slated to exhibit an even more immense growth of knowledge of the galaxy and its components: Projects like TESS and CHEOPS will find a slew of new exoplanets [15,16], with sensitivity pushing into the Earthlike regime. Experiments like the James Webb Space Telescope [17] are expected to directly characterize the atmospheres of several Earth-sized planets [18], the disequilibrium of which will make it possible to infer the presence or absence of biospheres [19]. In addition, further afield, when the next generation of telescopes such as TMT, PLATO, HabEx, and LUVOIR will be able to deliver a large enough population to do meaningful statistics on atmospheric properties [20], we will be able to characterize the ubiquity of microbial life, as well as which environmental factors its presence correlates with [21].

Rather than wait for the findings of these missions to further our understanding of the conditions for habitability, our position now represents a unique opportunity: We can test various habitability criteria for their compatibility within the multiverse framework. If a certain criterion is incompatible in the sense that it would make the values of the fundamental constants that we observe highly improbable among typical observers, then we can make the prediction that this criterion does not accurately reflect the habitability properties of our universe. When we are finally able to measure the distribution of life in our galaxy, if we indeed confirm this habitability criterion, then we will have strong evidence that the multiverse hypothesis is wrong. Likewise, if a criterion is necessary in order for our observations to be typical, but is later found to be wrong, this would be evidence against the multiverse as well. Put more succinctly, if our universe is good at something, we expect that to be important for life, and if it is bad at it, we expect it to not be important for life. Here, by saying that 'our universe is good at something', we really mean that by adopting the habitability criterion in question, our presence in this universe is probable, and, equally importantly, by not adopting it, our presence in this universe is improbable. Because it is easier to determine what our universe is good at than what life needs, the former can be done first, and used as a prediction for the latter. This logic is displayed in Figure 1.

A simple example will illustrate this approach: Suppose we take the hypothesis that the probability of the emergence of intelligent life around a star is proportional to its total lifetime. Then, in the multiverse setting, we would expect to live in a universe where the lifetime of stars is as long as possible. This is not the case, as we will show in [22]; therefore, the multiverse necessarily predicts that stellar lifetime cannot have a large impact on habitability. If, once we detect several biospheres, we find a correlation between the age of the star and the presence of life, we will have falsified this prediction the multiverse has made. Upcoming experiments aimed at characterizing the atmospheres of exoplanets will make this task feasible.

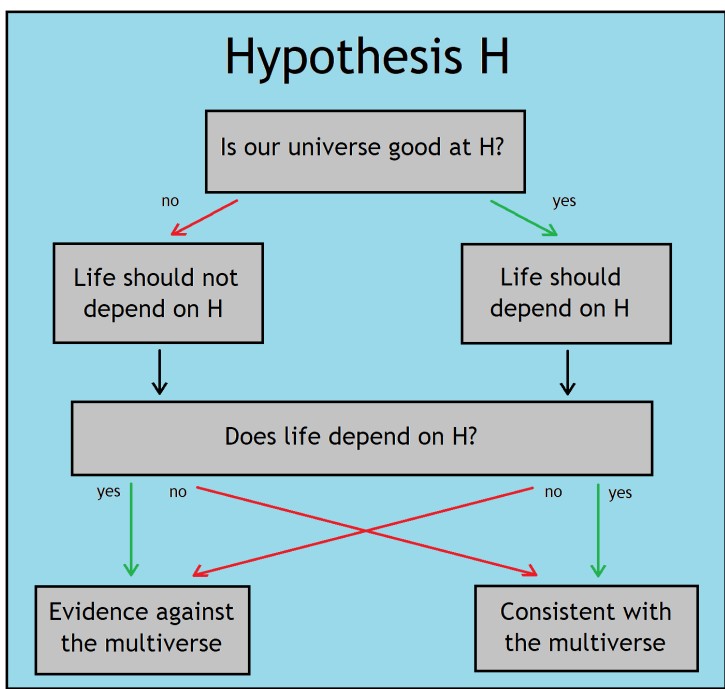

**Figure 1.** The underlying logic behind this task. We consider separate habitability hypotheses 'H', and determine whether our universe is good at H, in the sense that adopting this notion of habitability makes our presence in this universe probable (and equally importantly, that not adopting this notion makes our presence in this universe improbable). This yields a prediction for whether or not H is a good requirement for habitability that can then be tested against upcoming observations. There are dozens of proposed habitability criteria in the literature, and though not all of them will have a significant influence on our likelihood, many will. If we do live in a multiverse, we expect compatibility with all of these tests; if just one yields incompatible results, we will be able to rule out the multiverse hypothesis to a potentially very high degree of confidence.

This logic can be repeated for a number of different proposed habitability criteria: Estimates for the habitability of our universe change drastically depending on whether life can or cannot exist: On tidally locked planets; around dwarf stars; without the aid of photosynthesis; off of Earth-mass planets; outside the temperate zone; without plate tectonics; and in the presence of dangers such as comets, supernovae, and gamma ray bursts, among many others. Considering the impact of each of these can yield a separate prediction for what life requires, and consequently where we should expect to find it in upcoming surveys. Incorporating each will alter the distribution of observers throughout the purported multiverse, to differing levels of importance. Considering the gamut will include dozens of considerations for which should be crucial in the multiverse context, and promises to yield several strong predictions for where life should be found, sometimes up to a statistical significance of $6\sigma$. This undertaking will take some effort, but it promises to elevate the multiverse hypothesis to standard, falsifiable, scientific theory.

Let us further stress that each habitability criterion acts as a quasi-independent test, which can greatly strengthen our conclusions of whether the multiverse exists or not. There are arguments both for and against every criterion we consider here, and since the issue is not likely to be settled through logical argument, in general it would pay to remain agnostic toward which are true. The multiverse, however, makes specific predictions about which criteria are right. If even one of these predictions fails to be in accord with observations, we will have strong evidence against the multiverse hypothesis. Conversely, if all of the predictions we make are shown to be true, this would be strong evidence for the multiverse.

We initiate this task in this paper by considering only the simplest criteria for habitability, based off counting the number of potentially habitable stars. Section 2 is devoted to outlining the formalism

and detailing the observational facts that need to be reconciled. Section 3 presents the simplest possible estimate of habitability, where every star is taken as potentially life bearing with equal capability. We improve upon this in Section 4, where we compare multiverse expectations with the following proposed stellar habitability criteria: Is photosynthesis necessary for complex life? Is there a minimum timescale for developing intelligence? Are tidally locked planets habitable? Are convective stars habitable? Though most of the criteria we consider in isolation fail to yield a satisfactory account of our observed values, we finally display one that does, based off the total entropy produced throughout a star's lifetime, which serves as a minimal working model in terms of habitability criteria. We conclude in Section 5, and demonstrate that including multiple criteria simultaneously can lead to 'epistatic' effects in the probability of observing our parameters. Several concrete predictions for the distribution of life throughout our universe are made based off the success of each hypothesis, and adding further refinements to the scenarios considered here are capable of yielding multiple more.

## 2. Preliminaries

### 2.1. Properties and Probabilities of Our Universe

What are the properties of the world we are trying to explain with this approach? We focus here on three dimensionless constants, the fine structure constant $\alpha = e^2/(4\pi)$, where $e$ is the charge of the electron, the ratio of the electron mass to the proton mass $\beta = m_e/m_p$, and the strength of gravity $\gamma = m_p/M_{pl}$, where $M_{pl}$ is the reduced Planck mass. The values of these three quantities determine a great deal of the macroscopic characteristics of the universe [23]. In this approach, it is necessary to ask what values these parameters may take in order for the universe to be compatible with life, and where our observed values are situated within this allowed region. The different positions of these three variables will guide how sensitive we expect the criteria for habitability to be on these variables, which in turn will translate into an expectation for the types of environments that life is capable of thriving in.

The electron to proton mass ratio can be a factor of 2.15 larger than its current value before the processes involved in stellar fusion stop being operational [6], though detractors of this requirement [24,25] state that other fusion processes would take place. A factor of 4.9 and hydrogen would become unstable, creating a universe filled entirely with neutrons, incapable of complex chemistry (though even this scenario has been argued to be capable of producing life [26], highlighting the extreme degree of contention any speculative statements in this subject bring). Throughout this work, we will take the first, more stringent bound, as the border of the anthropically viable region, but relaxing this could readily be incorporated into our framework.

The fine structure constant also affects the stability of hydrogen, and will cause it to decay if it were 2.07 times larger. (It was found in [27] that a few percent increase would also preclude sufficient carbon production in stars, but this can be compensated by altering the pion mass). While this upper limit depends on the value of the electron to proton mass ratio, this can be compensated by the difference between the down and up quark masses, so that in effect the two limits are independent. A lower bound on the fine structure constant of about 1/5 the observed value was found in [5,28], based on the requirement for galactic cooling, which we will need to use as several scenarios we encounter favor small $\alpha$.

Unlike the other two quantities, the strength of gravity can be several orders of magnitude stronger without any adverse effect toward life. As we will show in Section 4, it can be roughly 134 times stronger before the lifetime of stars becomes shorter than biological evolutionary timescales. There are lower limits to this, as with all other quantities, as well, which we will delay discussing until [29], but the relevant point here is that none of these quantities are very closely situated to their minimum allowed values. Let us also comment on the bound found in [30], which appears to place a stronger upper bound on the strength of gravity than the one we use. The quantity they consider is $M_{pl}/\text{GeV}$, keeping particle physics fixed: As such, it relates to various cosmological processes that depend on the Planck mass. They determine that the rate of close encounters between star systems

is too frequent if this quantity is about ten times smaller, depending on the model for fluctuations, baryogenesis, and dark matter. While this is an important anthropic boundary that we will return to in the future, it can be alleviated by altering the other constants we consider, and turns out to be weaker than the stellar lifetime bound if $\alpha$ and $\beta$ are allowed to vary.

For a habitability criterion to be compatible with the multiverse, our measured values must be compatible with what a typical observer would find. In particular, the fact that the bounds on $\alpha$ and $\beta$ are only a few times larger than their observed values indicates that a relatively weak dependence on these quantities is preferred. In contrast, the habitability condition must impose a restriction that strongly favors weak gravitational strength in order to counteract the preference for larger values of $\gamma$.

Before we continue on, let us make this notion of typicality more quantitative. For each observable quantity $x = \alpha, \beta, \gamma$, we define the typicality $\mathbb{P}(x_{\text{obs}})$ as the cumulative probability that a value more extreme than ours is observed. Then, we have

$$\mathbb{P}(x_{\text{obs}}) = \min \left\{ \mathrm{P}(x > x_{\text{obs}}), \mathrm{P}(x < x_{\text{obs}}) \right\} \tag{1}$$

In this definition, we integrate over all other variables not under direct consideration, so that if there is a large portion of universes that have a different value of multiple parameters simultaneously, our observation is penalized. This combats the tendency for degeneracies in parameter space that we will encounter, which lead to misleading statistics if the other variables were to be held fixed. In addition, note that we take the minimum of the cumulative distribution function and its complement, to disfavor the scenarios where our observed values are anomalously close to the boundary of a region, though they may lie in a heavily favored location. With this definition, the maximal value of this quantity is 1/2.

It is also possible to define a global typicality, which counts the fraction of observers that would find themselves in universes less typical than ours: This is not quite reconstructible from the quantities above, but in all cases considered it yields no additional insights, so is not put to use here. Then, our statistic for whether a notion of habitability is compatible with the multiverse is the combination of the probabilities for the three parameters parameters we consider, $\alpha$, $\beta$, and $\gamma$.

*2.2. Drake Parameters*

Now that the expectations for the definition of habitability have been outlined, we must find a way to estimate the relative number of observers within a universe, and then extract how this quantity depends on the underlying physical parameters. Fortunately, the technology for doing this has been developed some time ago, as the well known Drake equation. Here, we make use of a slight modification of the 'archaeological form' of the Drake equation outlined in [31]: Our expression for the habitability of a given universe is equal to the expected number of observers that are produced throughout the course of its evolution. This can be broken down into the following product of factors:

$$\mathbb{H} = N_\star \times f_{\text{p}} \times n_{\text{e}} \times f_{\text{bio}} \times f_{\text{int}} \times N_{\text{obs}} \tag{2}$$

where here $N_\star$ is the number of habitable stars in the universe, $f_{\text{p}}$ is the fraction of star systems that contain planets, $n_{\text{e}}$ is the average number of habitable planets in systems that do possess planets, $f_{\text{bio}}$ is the fraction of habitable planets on which life emerges, $f_{\text{int}}$ is the fraction of life bearing planets that ultimately develop intelligent organisms, and $N_{\text{obs}}$ is the number of observers per intelligent species. The first application of the Drake equation to multiverse reasoning was in [32].

As always, the point of this equation is meant to be organizational: It marshals the great variety of factors that dictate the emergence of intelligence into largely factorizable subproblems, and allows us to cleanly isolate the assumptions that go into each. While, as usual, the overall normalization remains highly uncertain, use of this equation will allow us to directly compare the relative numbers of observers for any two universes, given that we state our assumptions on how each of these parameters depends on the laws of physics.

Each factor in this equation deserves a fair amount of attention in its own right, and consequently, rather than overload the reader with every consideration that goes into this analysis at once, we will split this estimation into multiple separate papers. Our strategy will be to work our way through the Drake factors one at a time, starting from the left and moving our way to the right. It is important to note that although the final conclusions can only be made once all of these factors are considered in a unified picture, care is taken to report only those results that carry through once this synthesis takes place, and to explicitly state when conclusions will be altered in the full analysis. A completely satisfactory account of all three of our observed values will only be achieved once $f_{bio}$ is taken into the fold.

We begin, then, in this paper, by considering how the number of habitable stars depends on the laws of physics, effectively taking the simplified ansatz that the number of observers will be directly proportional to the number of stars, independent of any other properties of the universe. We will first define precisely what is meant by this quantity in an infinite universe, then use straightforward estimates for the average size of stars to arrive at our first, simplest potential definition of habitability. This will be shown to be incompatible with the multiverse hypothesis, which motivates searching for refinements to improve. Next, we discuss additional proposed criteria for stellar habitability, and compare how these criteria fare.

Subsequent papers will deal with the other factors. In [29] we will discuss the two relating to the formation of planets. We will confront which factors are necessary for a star to produce planets, their resultant properties, and what the conditions on the parameters are required in order to achieve this. We will find that these considerations do not alter the probability distributions themselves much, but that they do place strong bounds on the allowed parameter space, many of which are the strongest lower bounds that can be found in the literature.

Next, in [22] we discuss what planetary characteristics may possibly influence the advent of simple life. We will consider a slew of possibilities here, and find that most of them are incompatible with the multiverse hypothesis, leading to clear predictions for where life should be found in our universe.

In [33] we tackle the question of how often intelligence emerges from simple life. Our approach will be to determine the rate of suppression of intelligent life, such as the mass extinctions that have plagued our planet over the course of geological time, and how the rates of these depend on the physical constants. Throughout, we will emphasize the testable predictions the multiverse hypothesis offers, and suggest the quickest ways to falsify them.

Before we begin, we need to relate the habitability to the probability of measuring particular values of the observables, because they need not be exactly equal: If there is an underlying prior distribution of the space of variables, presumably set by the ultimate physical theory, this must be taken into account as well, so that the probability of finding oneself in a given universe will be proportional to the habitability of that universe multiplied by that universe's chances of occurring:

$$\mathrm{P}\left(\alpha,\beta,\gamma\right) \propto \mathbb{H}\left(\alpha,\beta,\gamma\right)\, p_{\mathrm{prior}}\left(\alpha,\beta,\gamma\right) \tag{3}$$

While the precise form of the prior may need to await a fuller understanding of the ultimate theory of nature, we can make a plausible ansatz for each of the variables we are concerned with here. As we will find, the habitability often depends on the parameters much more strongly than the prior anyway, and so adopting a mildly different one will not appreciably alter any of our results.

We expect the prior on the fine structure constant $\alpha$ to be nearly flat, without any strong features or special values that would skew the distribution too much in their favor. Again, if the reader has reason to believe in some other prior, the details will not change much. The ratio of proton mass to Planck mass, on the other hand, is taken to be scale invariant, or flat in logarithmic space: $p_{\mathrm{prior}}(\gamma) \propto 1/\gamma$. The reasoning behind this is that the proton mass is dictated by the scale at which the strong force becomes confining, which through renormalization group analysis is given by $m_p \sim M e^{C-2\pi/(9\alpha_s)}$, where $M$ is some large mass scale, $\alpha_s$ is the strength of the strong force at high energies, and $C$ is a coefficient that depends on the heavy quark masses. If this is taken to be roughly uniform at that scale, then the

distribution for $\gamma$ will be scale invariant (up to unimportant logarithmic corrections, which anyway depend on the precise distribution for the coupling). Similarly, the ratio of electron mass to proton mass will also be taken to be scale invariant, since not only is the proton mass scale invariant, but there is also reason to suspect that the Yukawa couplings of the standard model follow a scale invariant distribution as well [34]. Then, our final result for the probability of measuring a particular value of the parameters will be:

$$P(\alpha, \beta, \gamma) \propto \frac{\mathbb{H}(\alpha, \beta, \gamma)}{\beta\,\gamma} \tag{4}$$

Note that adopting this distribution, based off the plausibility of high energy physics priors, assumes that these are uncorrelated with the properties which affect how likely a particular universe is to arise, such as the reheating temperature, or the nucleation rate in the context of false vacuum inflation. This is a plausible assumption, as these properties of the theory are presumably set by the inflaton sector, that is insensitive to the masses and couplings of light states, but if the reader has reason to suspect an alternative scenario they should feel free to adopt their own prior. In addition, note that in this work we assume that the multiverse context accommodates all three of these quantities as variable and uncorrelated. This also seems plausible, especially since two of them are composed of multiple factors, but an alternative view could readily be incorporated in this formalism as well. We do not consider universes that are radically different than our own, such as having different types of particles, forces, or number of dimensions. We regard the comparison with potential habitats in those universes as too speculative to make immediate progress, and anyway not amenable to the type of reasoning we employ here.

With this, we are now ready to make our simplest appraisal of the overall habitability of our universe.

## 3. Number of Stars in the Universe $N_\star$

### 3.1. What Is Meant by this Quantity?

In this section we elaborate on what we mean by the number of stars in a given universe. If universes are infinite, this is an ill-defined concept, and comparing the relative number of stars in two different universes is ambiguous, which is a manifestation of the measure problem [35,36].

The fact that there is no obvious unique choice for this comparison has plagued cosmologists since the early days of multiverse reasoning. Many proposals have been made for how to regulate the infinities one must deal with, in order to be able to compare two finite numbers. Most proposals immediately run into drastic conflict with observation, which helps to winnow down the possibilities to a smaller subset. Encouragingly, of the few that remain, several have been shown to be equivalent to each other, even though the starting points were radically different [37,38]. However, as it stands there are multiple existing measures, with no obvious way of specifying which is correct. Thankfully for our purposes, much of this ambiguity only affects the distribution of cosmological parameters, leaving the microphysical parameters that we focus on relatively independent of the measure.

Here we make use of the scale factor cutoff measure [39], which states that the probability of an observer arising in a particular universe should be simply proportional to the number of baryons that have found their way into a suitably large galaxy cluster by the time the universe has reached a certain size. This cutoff size is arbitrary, and will not affect probabilities as long as it is taken to be longer than the time of peak galactic assembly. Additionally, the total number of baryons is infinite in an infinite universe, motivating the need to regulate by truncating to a finite region of space: In practice this can be accomplished by merely noting that the ratio of two probabilities is then equal to the ratio of baryon densities, again independent of cutoff. (Here, care must be taken to regularize in an unbiased way, since densities change in expanding universes, but as noted this will only have an affect cosmological parameters).

While the requirement on cluster mass is usually used to penalize universes that do not produce large enough halos (for instance, if the cosmological constant is too large [2] or the density contrast too small [3,5]), the reason for this has to do with the formation with planets, and so will not concern us here. We will return to this subject in [29], where we investigate what sets the minimum mass of a halo in terms of fundamental parameters. For now, we restrict our attention to universes where the majority of baryons falls into star forming regions, and instead add a layer of sophistication to the criteria, that the number of observers produced will be proportional to the number of stars.

To begin, we make the simplification that the efficiency of star formation, that is, the total amount of matter that ultimately becomes stars, is independent of the halo mass, and equal to $\epsilon_\star = 0.03$ [40]. This is in fact not a good approximation, as this quantity is known to be affected by many feedback processes that can lower the efficiency for both small mass and large mass halos [41]. We will refine our prescription to take these effects into account in a future publication, but expect that they should play more of a role for cosmological observables, rather than the ones we focus on here. Additionally, the efficiency is not taken to be a strong function of microphysical parameters. (Though it does in principle: For example, if the fine structure constant is too low, cooling will be so inefficient that gas will never fragment into collapsing clouds [5].)

### 3.2. Is Habitability Simply Proportional to the Number of Stars?

We are now ready to make our simplest estimate of the habitability of a universe, that the number of observers is proportional to the number of stars. This makes the quite unreasonable assumption that every star is equally habitable, yet it will serve as the calculational substrate, on top of which further refinements can be added. Modifications of this basic framework are the subject of the later sections of this paper, as well as the forthcoming sequels. Then, it stands to reason that the number of stars will be inversely proportional to how large they are: A universe where stars are smaller would ultimately be able to make more of them with the same amount of initial material. If each star is an independent opportunity to develop intelligent life, then universes that produce the smallest possible stars would have the greatest number of observers.

How large are stars, then? As is well known, the typical stellar mass scale is given by the quantity $M_0 = (8\pi)^{3/2} M_{pl}^3/m_p^2 = 1.8 M_\odot$ (e.g., [42]). However, the average stellar mass is considerably lighter than this, and exhibits additional dependence on the physical constants. To estimate the average, we need to know the distribution of stellar masses. This is given by the initial mass function (IMF); for the majority of this paper we take this to be of the classic power law form, $p_{\text{IMF}}(\lambda) \propto \lambda^{-\beta_{\text{IMF}}}$ [43], which defines the dimensionless quantity $\lambda = M_\star/M_0$. The quantity $\beta_{\text{IMF}} = 2.35$ is referred to as the Salpeter slope, and, as usual with power law exponents, is set by universal processes that do not depend on physical parameters [44]. More realistic treatments instead use a broken power law, lognormal distribution, or some other form [45,46], which accurately reflects the details of the feedback mechanisms accompanying star formation, but this is an unnecessary complication that obfuscates but does not appreciably change our results. We will incorporate a more sophisticated IMF into this formalism in Section 5. The most relevant feature of this distribution is that it is very steep, making the vast majority of stars born rather close to the minimal possible mass, and larger stars extremely rare. As such, the average stellar mass is simply proportional to the minimum, and so it will be essential to estimate this quantity.

The minimum stellar mass can be determined based off the requirement that its central temperature must be high enough to ignite hydrogen fusion. Particular attention was paid to the dependence of this minimum mass on physical parameters in [42], where the central temperature of a star was determined to be $T \approx \lambda^{4/3} m_e$. This must be compared to the required temperature, which in [47] was found to be given by the Gamow energy, the threshold above which thermal fluctuations can routinely instigate tunneling through the repulsive barrier between two protons, $T_G \sim \alpha^2 m_p$. Demanding the central temperature be greater than this gives $\lambda_{\text{min}} = 0.22 \alpha^{3/2} \beta^{-3/4}$. This same scaling

was also found in [23], where they demanded the essentially equivalent requirement that the scattering time is shorter than the Kelvin-Helmholtz time.

The average stellar mass can be computed from the initial mass function we employ as $\langle \lambda \rangle = (\beta_{\mathrm{IMF}} - 1)/(\beta_{\mathrm{IMF}} - 2)\lambda_{\min}$, but this actually underestimates the average stellar mass. The origin of this discrepancy come from the fact that the initial mass function deviates from a power law for small masses, reflecting feedback in the star formation process [44]. However, the normalization of this value is not important for our analysis, since it will only enter into the probability multiplicatively, and so we can simply take $\langle \lambda \rangle \propto \lambda_{\min}$. This relation holds in more realistic treatments as well.

A maximum stellar mass also exists, based on the criterion that a star must be gravitationally stable. This was found to be $\lambda_{\max} = 56$ in [42], independent of any constants. This cutoff may easily be included in our analysis, but it would considerably complicate the final expressions for the probability. Due to the extreme rarity of stars larger than this mass, we neglect this cutoff here, which does not alter any of the numbers we find to the precision we report.

The expected habitability of a universe with given constants is then:

$$\mathbb{H}_\star \propto N_\star \propto \frac{\epsilon_\star \, m_p}{\langle \lambda \rangle \, M_0} \propto \frac{\beta^{3/4} \gamma^3}{\alpha^{3/2}} \tag{5}$$

This, along with the measure from Equation (4), determines the probability distribution for observing values of the three parameters under consideration. Bearing in mind the total range for these values, we may calculate the probabilities of observing ours to be:[1]

$$\mathbb{P}(\alpha_{\mathrm{obs}}) = 0.20, \quad \mathbb{P}(\beta_{\mathrm{obs}}) = 0.44, \quad \mathbb{P}(\gamma_{\mathrm{obs}}) = 4.2 \times 10^{-7} \tag{6}$$

The number of habitable stars is plotted in three different subplanes of the parameter space in Figure 2[2].

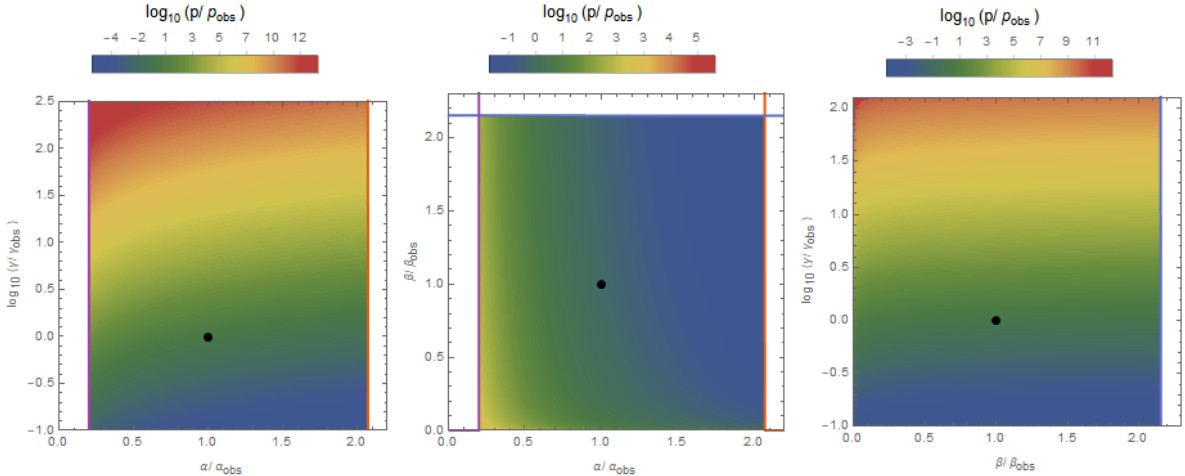

**Figure 2.** The distribution of stars throughout the multiverse. Each point represents a universe with a given set of parameters; the black dot represents our values. A strong preference for large $\gamma$, weak preference for small $\beta$, and a slightly stronger preference for small $\alpha$ value can be seen. Note that the $\gamma$ axes are logarithmic, as well as the color display for the probability, which spans 16 order of magnitude.

---

[1]  The code to compute all probabilities discussed in the text is made available at https://github.com/mccsandora/Multiverse-Habitability-Handler.

[2]  The above used the log-uniform prior for $\beta$ and $\gamma$, as discussed above. If instead we use a uniform prior, we find $\mathbb{P}(\alpha_{\mathrm{obs}}) = 0.20$, $\mathbb{P}(\beta_{\mathrm{obs}}) = 0.26$, and $\mathbb{P}(\gamma_{\mathrm{obs}}) = 3.1 \times 10^{-9}$. We see that the probabilities for $\alpha$ and $\beta$ are affected only slightly, and the probability for $\gamma$ is decreased by two orders of magnitude. This is a fairly typical result.

As can be seen, there are allowed values of the parameters that contain many more stars than in our universe. Most notable is the preferences for larger gravitational strength, which represents a $5.1\sigma$ deviation from typicality. Based off this consideration alone, we conclude that this habitability criterion is incompatible with the premise that we are typical observers in a multiverse ensemble. This allows us to make our first testable prediction, that not all stars in our universe should be equally habitable. This should strike the reader as somewhat of an underwhelming prediction, since there are few that would wager against this statement, but it demonstrates a first possible test of the multiverse hypothesis. Including subsequent layers of realism will yield concomitantly sophisticated predictions.

## 4. Habitability Dependent on Stellar Properties

Up to this point, we have treated every stellar mass as potentially habitable, and postulated that the universe should maximize the total number of stars, irrespective of their properties. This is not justified, as there are a number of criteria that could render a star incapable of supporting life-bearing planets, and so now we develop the tools to reflect that in our calculations. Here, we focus on stellar characteristics that are a function of the mass only, though other aspects, such as metallicity, will be dealt with in [29]. Even further aspects, such as rotation, composition, environment, etc. would be interesting to investigate in the future. We first consider each additional criterion in isolation in this section, and then in Section 5 we consider various combinations of criteria to investigate their joint effects.

### 4.1. Is Photosynthesis Necessary for Complex Life?

Photosynthesis, the process by which photons are harvested by life for the purposes of creating chemical energy, was one of the absolutely key innovations in the history of life on our planet. By using the energy imparted on a specific molecule, this mechanism makes it possible to strip off electrons, which can then be used to process carbon dioxide into sugar. While other sources of energy may be exploited for this task [48], the sheer magnitude of available energy coming from the sun makes the harvesting of this source unrivaled, 3 orders of magnitude above any other potential source [49]. Today, there are many different molecular bases for anoxygenic photochemical systems, suggesting that it arose independently many times [50] and that it arose quite early in the history of the planet, perhaps 3.5 Ga [51], or even 3.8 Ga [52]. Photosynthesis provides the basis for the organic material for essentially the entire biosphere today (even at hydrothermal vents, who utilize organic material precipitated from above [48]). It was argued in [48] to inevitably arise in any situation where organic carbon is scarce and light energy is available.

Oxygenic photosynthesis is even more crucial for life on Earth. The ability to harvest the hydrogen atoms from water molecules allowed the process to yield 18 times the amount of energy as anoxygenic photosynthesis [53,54], provided a much more abundant supply chain, and, subsequently, oxidized the entire atmosphere. The Cambrian explosion occurred only after this event, and many complex organisms, including ourselves, require high levels of oxygen to perform the necessary level of metabolic activity [55]. Additionally, the atmospheric oxygen content was essential for the development of an ozone shield, which allowed subsequent colonization of the land surface.

Photosynthesis is by no means automatic, however. It relies on a coincidence where the energy of photons produced by the sun is roughly coincident with the energy required to ionize common molecules. This fact, that starlight is right at the molecular bond threshold, is one of the most remarkable anthropic coincidences. Originally pointed out by [56] on the basis that the stellar temperature be such that molecular bonds may form a partially convective outer layer, this bound was reinterpreted by [23] by noting that the energy may be harvested for chemical purposes. The requirements on the fundamental parameters can be found by equating the surface temperature of a star with the Rydberg energy. Using formulas in the appendix, this can be seen to occur for sunlike stars only if:

.

$$\alpha^6 \left( \frac{m_e}{m_p} \right)^2 \approx \frac{m_p}{\sqrt{8\pi}\, M_{pl}} \tag{7}$$

In [57] the precise details of the star were taken into account more carefully, altering the form of this expression slightly depending on the type of scattering that occurs. It is striking how well this equality holds in our universe, where the two are equal up to a factor of 1.7. Here, the temperature dependence on the size of the star is not taken into account in this expression because the spread in stellar temperatures is actually quite small. However, this makes the degree of tuning somewhat obtuse, and so we improve upon this standard analysis in what follows. This motivates our second ansatz for the habitability of a universe, that the number of observers is proportional to the number of stars capable of eliciting photosynthesis.

This begs the question, of what the allowable range for photosynthesis actually is. Though evidence across many different lineages suggests that photosystems have evolved to utilize the wavelengths with the most number of photons, (subject to some additional considerations) [54], there are hard physical limits for which wavelength photons are potentially photosynthetically useful. A lower bound often quoted is 400 nm [58] as below this photodissociation of most molecules occurs (though fluorescent pigments may potentially circumvent this bound [59]). An upper bound of 1100 nm was deduced in [60] on the basis that below this energy photons are indistinguishable from vibrational modes of molecules. A species of purple bacteria has been found that utilizes 1020 nm photons, though to split electrons from ferrous iron, which requires less energy [54]. The longest wavelength used for splitting water was recently found to be 750 nm [61]. In the following, we will refer to the maximally optimistic wavelength range, between 400–1100 nm, as the 'photosynthesis criterion', and the range 600–750 nm as the 'yellow criterion'. In this section we will stick to the former, and in the following subsection vary these bounds, commenting on the implications for what locales photosynthesis should be found around throughout our universe.

With this condition, the habitability of a universe becomes $\mathbb{H} = N_\star f_{\mathrm{photo}}$, with:

$$f_{\mathrm{photo}} = \int_{\lambda_{\mathrm{fizzle}}}^{\lambda_{\mathrm{fry}}} d\lambda \; p_{\mathrm{IMF}}(\lambda) \tag{8}$$

Here $\lambda_{\mathrm{fizzle}}$ is the stellar mass with spectral temperature too weak for photosynthesis, and $\lambda_{\mathrm{fry}}$ the mass which is too hot. These both depend on the values taken for the limiting wavelengths, as illustrated in Figure 3. This should give the reader some idea of the width of allowed values of the parameters, which was not reported in the original treatments of this coincidence.

This leads to the following estimate for the habitability of the universe:

$$\mathbb{H}_{\mathrm{photo}} \propto \alpha^{-3/2}\, \beta^{3/4}\, \gamma^3 \left( \min\left\{ 1, 0.45 \frac{L_{\mathrm{fizzle}}}{1100\,\mathrm{nm}} Y^{1/4} \right\}^{2.84} - \min\left\{ 1, 0.16 \frac{L_{\mathrm{fry}}}{400\,\mathrm{nm}} Y^{1/4} \right\}^{2.84} \right) \tag{9}$$

With $L_{\mathrm{fizzle}}$ and $L_{\mathrm{fry}}$ being the longest and shortest suitable wavelengths, respectively, and:

$$Y = 3.19 \frac{\gamma}{\alpha^{63/20}\, \beta^{137/40}} \tag{10}$$

which is normalized to 1 for our observed values. As can be seen, this quantity, which controls the fraction of photosynthetic stars, differs from the expectation given by Equation (7). This discrepancy is due to the fact that the original analysis restricted attention to sunlike stars, whereas we have considered the entire range of stars as potentially photosynthetic. This criterion leads to probabilities:

$$\mathbb{P}(\alpha_{\mathrm{obs}}) = 0.32, \quad \mathbb{P}(\beta_{\mathrm{obs}}) = 0.23, \quad \mathbb{P}(\gamma_{\mathrm{obs}}) = 5.2 \times 10^{-7} \tag{11}$$

When compared with Equation (6), the typicality of our observed electron to proton mass ratio is worse by about a factor of 1.9, the fine structure constant better by 1.6, and the strength of gravity better by 1.3. So far, this hypothesis does not seem to add much to the discussion. However, it is premature to dismiss it: As can be seen from Figure 4, its main effect is to enforce a somewhat tight relationship between the parameters $\alpha$ and $\beta$, but it retains the tendency to prefer large values of $\gamma$. When used in conjunction with additional criteria to be discussed below, this will become an essential ingredient in finding a definition of habitability that renders all three probabilities very likely.

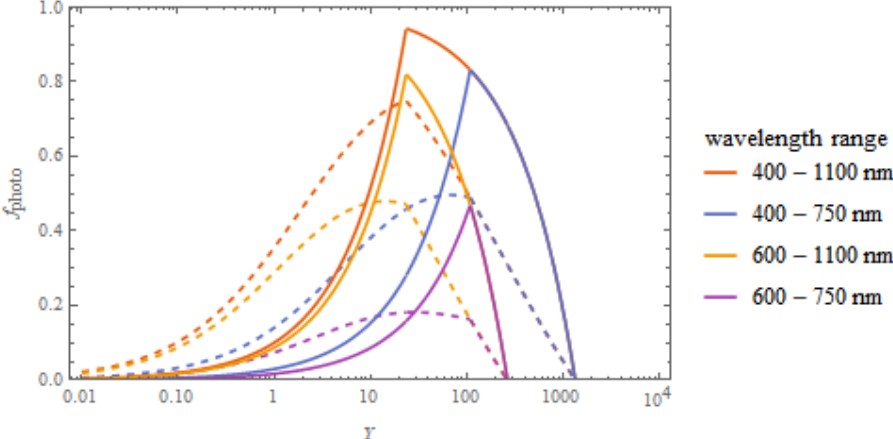

**Figure 3.** The fraction of stars which are capable of supporting photosynthesis as a function of the composite parameter $Y$ defined in the text. The different curves correspond to taking the minimal wavelength to be both 400 nm and 600 nm, and the maximal to be 750 and 1100 nm. The solid curves use the estimate in Equation (8), and the dashed curves use the more refined initial mass function (IMF) of Equation (26).

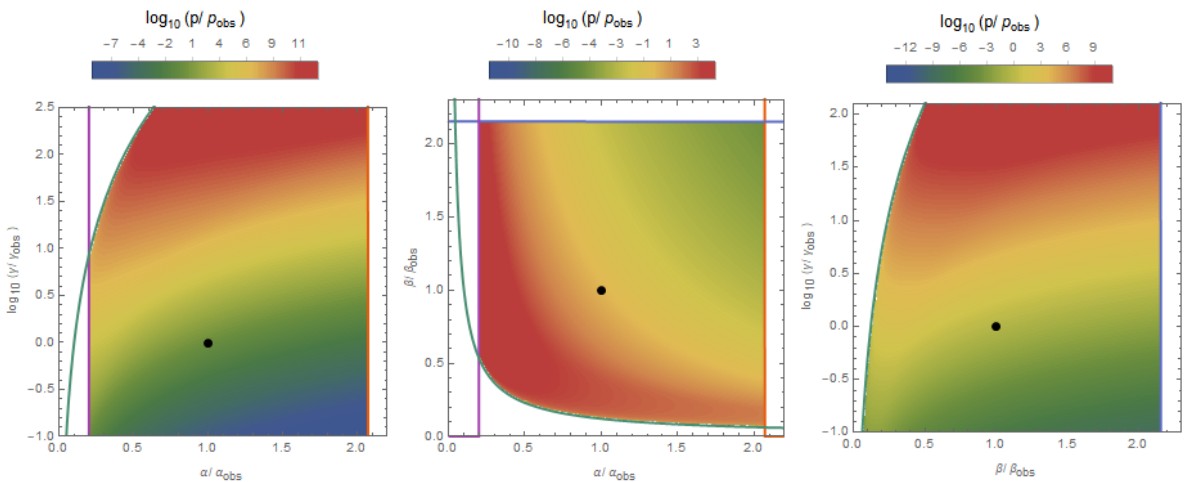

**Figure 4.** Distribution of observers from imposing the photosynthesis condition. A strong preference for the parameters to be restricted to the photosynthetic range is introduced, but there is still a preference for large $\gamma$.

Before moving on, there is also a lower limit on flux needed to support photosynthesis: This is commonly taken to be 1% of the average surface flux on Earth. (Though an organism has been found that subsists on $10^{-5}$ of this level, it would be incapable of sustaining the biosphere as we know it) [62]. The origin of this lower bound is that at least one photon should be incident on the photosystem molecules per cycling time, which from the appendix gives $\Phi_{\text{min}} \sim T_{\text{mol}}^3 = 5 \times 10^{-5} \alpha^6 m_e^{9/2}/m_p^{3/2}$, in excellent accord with the actual value. Comparing this to the flux from sunlike stars gives the

following bound on the physical constants: $\alpha^{3/2}\beta\gamma^{-1/4} > 1.4 \times 10^{-4}$. This is a rather mild criterion, and will not be anthropically relevant. This would be useful in a more sophisticated analysis that determines the potentially photosynthetic stars by considering when the useful part of their spectrum falls below this threshold along the lines of [63], rather than our somewhat simplified prescription of scaling based off the surface temperature. We do not expect this more elaborate treatment to substantially alter our conclusions.

### 4.2. Is Photosynthesis Possible around Red Dwarfs?

Without understanding the extent of the range of wavelengths capable of giving rise to photosynthesis, there is some ambiguity to the amount of tuning that is required for it to hold. Additionally, we may wonder how our assumptions about the minimum or maximum wavelength thresholds affect the probabilities of our observed quantities, and whether these considerations can be used to inform our expectations of where photosynthesis can arise. We address these points here.

To split a hydrogen off a water molecule, it takes 1.23 eV [64]. In order to perform this splitting, it is necessary for life to utilize two of the sun's photons, at 680 nm and 700 nm (1.78 eV and 1.82 eV) [54], to perform a concatenated cascade of excitations known as the Z process.

If we assume that the maximal efficiency of oxygenic photosynthesis is $\epsilon = E_{H_2O}/E_{tot}$, which is equal to $\epsilon = 0.33$ on Earth, and also that photosynthesis can occur by the collection of $n$ photons per molecular bond (2 on Earth, as per our counting), then the longest wavelength possible is $\lambda_{fizzle} = \epsilon\, n\, 1008$ nm. The complicated nature of this two stage process likely delayed the evolution of oxygenic photosynthesis considerably, despite its great advantages [65].

On this basis, it was argued in [53] that photosynthesis could in principle take place around red dwarf stars, though it would take three photons per water splitting, and so may be proportionately harder to evolve. Additionally, it was found in [66] that it may be less productive, depending on the wavelength of light harvested. Ref. [58] argue that photosynthesis may take place around potentially F, G, K, and M star types, and [67] argue that even brown dwarfs and black smokers may support photosynthesis of some type.

In the above above analysis, we effectively assumed that photosynthesis will evolve in the entire physically allowed wavelength range. We may consider how our analysis changes with different assumptions, however. A few representative values are considered in Table 1, to illustrate the change in the probabilities that is effected. We can see that there is not a large difference introduced, so that the multiverse has little to say about whether we should expect photosynthesis around red dwarf stars in this regard. However, it will be useful to keep such effects in mind for future purposes, when we consider our location within this universe as well.

**Table 1.** Dependence of the probabilities of our observed quantities on the upper and lower limits of the photosynthetic range.

| Wavelength Range | $\mathbb{P}(\alpha_{obs})$ | $\mathbb{P}(\beta_{obs})$ | $\mathbb{P}(\gamma_{obs})$ |
|---|---|---|---|
| 400–1100 nm | 0.318 | 0.231 | 5.18e-07 |
| 400–750 nm | 0.242 | 0.263 | 3.85e-07 |
| 600–1100 nm | 0.444 | 0.175 | 7.35e-07 |
| 600–750 nm | 0.334 | 0.221 | 5.40e-07 |

### 4.3. Is There a Minimum Timescale for Developing Intelligence?

Until this point, we have not specified any bound that places an upper limit on the strength of gravity, and so the probabilities we have reported can be viewed as optimistic(!) estimates. However, our treatment has disregarded any mention of the actual lifetimes of the stars considered, which should surely influence the habitability properties of their surrounding planets. Here, we rectify this. The purpose is not to fully investigate the influence of stellar lifetime on habitability, which will be treated

more fully in [22], but rather to investigate the potential importance of this restriction. As a byproduct, we will find a maximal value of $\gamma$ for use throughout our calculations.

This can be done by noting that there is a maximum allowable mass for a habitable star, based on the criterion that it last long enough for life to take hold. Here, we make the crude approximation that all stars above this mass are inhospitable, and all below are equally habitable. How to define the timescale necessary for life is very uncertain—here we simply take it to be $t_{\text{bio}} = N_{\text{bio}} t_{\text{mol}}$, with $N_{\text{bio}} \sim 10^{30}$ and, $t_{\text{mol}}$ is the molecular timescale given in the appendix. For our universe, this should be on the gigayear timescale, an estimate suggested in [68]. This subscribes to the notion that this amount of time is both necessary and sufficient for a biosphere to achieve the complexity we observe, as advocated for example in such papers as [69]. Alternative viewpoints are certainly taken on this matter, which makes the adoption of this criterion that of a personal preference at the moment. These alternatives will be explored fully in [22].

Then, using the formula for stellar lifetime in the appendix, the maximal mass is:

$$\lambda_{\text{bio}} = 1.8 \times 10^{-13} \, \alpha^{8/5} \, \beta^{-1/5} \, \gamma^{-4/5} \tag{12}$$

The normalization has been set to match that observed in our universe, $\lambda_{bio} = 1.2$ [70], corresponding to the largest stars that last 1 Gyr, around 2 solar masses.

Ensuring that this maximum mass is larger than the minimum stellar mass yields an upper bound for $\gamma$ and unimportant lower bounds for $\alpha$ and $\beta$. The global upper bound on $\gamma$ is found to be $\gamma_{\text{max}} = 134$, the value implicitly used in all calculations above.

The habitability can then be expressed as:

$$\mathbb{H}_{\text{bio}} = N_\star f_{\text{bio}} \propto \alpha^{-3/2} \beta^{3/4} \gamma^3 \left( 1 - \min\left\{ 1, 1.48 \times 10^{15} \, \alpha^{-0.14} \beta^{-0.74} \gamma^{1.08} \right\} \right) \tag{13}$$

This is displayed in Figure 5. The most interesting feature that this criterion entails is the fact that the maximum value of $\gamma$ is now dependent on $\alpha$ to some extent, and especially $\beta$. This gives rise to a preference for larger values of the latter quantity by virtue of there being more observers for larger $\gamma$. This effect serves to make the probability of our observed value of $\beta$ less likely, as follows:

$$\mathbb{P}(\alpha_{\text{obs}}) = 0.28, \quad \mathbb{P}(\beta_{\text{obs}}) = 0.12, \quad \mathbb{P}(\gamma_{\text{obs}}) = 9.5 \times 10^{-6} \tag{14}$$

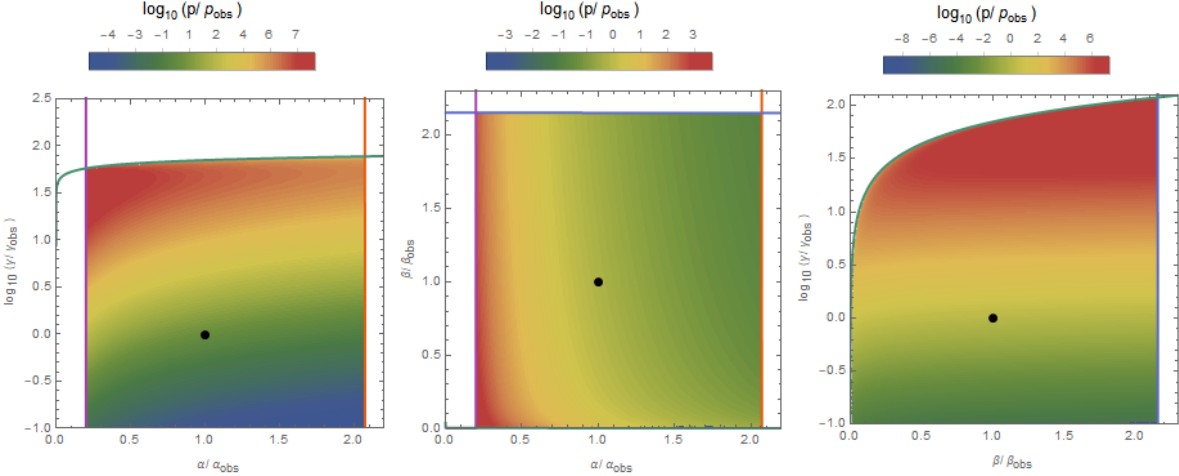

**Figure 5.** Distribution of observers from imposing the biological timescale condition. Of note is the secondary preference for large $\beta$ the anthropic boundary induces.

### 4.4. Are Tidally Locked Planets Habitable?

Traditionally, it has been argued that life would have a very hard time evolving on tidally locked planets (for a review see [71]): One side is perpetually scorched by its host star, and the other eternally shrouded in frigid darkness. Winds between the two hemispheres would tear over the surface, and rotation would be too slow to generate a magnetic shield. However, some researchers feel this dismissiveness may be unjustified: Recent climate modeling that suggests even a thin atmosphere would serve to taper these extreme conditions [72]. Even still, a recent paper [73] argues that stratospheric circulation may not be as clement on worlds like these after all. Clearly, it is premature to think that we know enough about all the complex processes on these worlds to be able to definitively conclude whether they may be potentially habitable or not, and new surprises are sure to abound. In this section we investigate how the estimates for the number of observers in a universe would change if we assume that tidally locked stars are not habitable.

If we adopt this, it will define a minimum allowable mass, as planets around smaller stars must orbit much closer to remain inside the temperate zone[3]. For this we use the standard formula for the time it takes a planet of mass $M$, radius $R$, and distance $a$ away from its mass $M_\star$ star to spin down from initial angular frequency $\omega$ [74]:

$$t_{\text{TL}} = \frac{40}{3} \frac{\omega\, a^6\, M}{G\, M_\star^2\, R^3} \tag{15}$$

The coefficient $40/3$ assumes the planet is rocky and roughly spherical. When planets are formed, they are nearly marginally bound: This sets the initial centrifugal force to be approximately equal to the force of gravity at the planet's surface, yielding $\omega \sim \sqrt{G\rho}$. For the Earth this gives a period of roughly 3 h, about twice as fast as our planet's initial rotation speed.

Using these expressions, the tidal locking time can be expressed as:

$$t_{\text{TL}} \sim 566 \frac{\lambda^{17/2}\, m_p^{17/2}}{\alpha^{51/2}\, m_e^{15/2}\, M_{pl}^2} \tag{16}$$

Note however the high powers involved, indicating that tidal locking is extremely sensitive to stellar mass.

We now have to compare this to the total habitable lifetime as a function of mass. Because stars steadily increase in luminosity as they age, the habitable zone migrates outwards, eventually causing even ideally situated planets to boil over. The habitable time of a star can then be defined as the average amount of time its orbits stay within the habitable zone. This requires knowledge of how quickly the star's luminosity changes, but the end result is just an order one factor of the total stellar lifetime, $t_{\text{hab}} = 0.4 t_\star$, independent of mass, and, more importantly, independent (or only very weakly dependent) on the physical parameters. The condition that $t_{\text{TL}} > t_\star$ gives $\lambda > \lambda_{\text{TL}}$, where:

$$\lambda_{\text{TL}} = 0.89\, \alpha^{5/2}\, \beta^{1/2}\, \gamma^{-4/11} \tag{17}$$

---

[3]　We entertain adopting a different stance as to whether planets must be in the temperate zone to be habitable in [29].

Here, the coefficient is set to agree with the estimate $\lambda > 0.47$ $(0.85 M_\odot)$ [75] which was found to be the threshold mass in our universe[4]. This can then be used in Equation (4) to yield:

$$\mathbb{H}_{\text{TL}} = N_\star \, f_{\text{TL}} \propto \alpha^{-3/2} \, \beta^{3/4} \, \gamma^3 \, \min\left\{1, 0.154 \, \alpha^{-1.35} \, \beta^{-1.69} \, \gamma^{0.49}\right\} \tag{18}$$

This gives probabilities of our observed values as:

$$\mathbb{P}(\alpha_{\text{obs}}) = 0.12, \quad \mathbb{P}(\beta_{\text{obs}}) = 0.30, \quad \mathbb{P}(\gamma_{\text{obs}}) = 1.8 \times 10^{-7}, \tag{19}$$

The distributions are visualized in Figure 6. Adding this consideration does not appreciably alter these probabilities, amounting to a factor of 1/6 amongst all three. However, the distribution looks markedly different: There is a much steeper dependence on $\alpha$ and $\beta$, favoring smaller values for each.

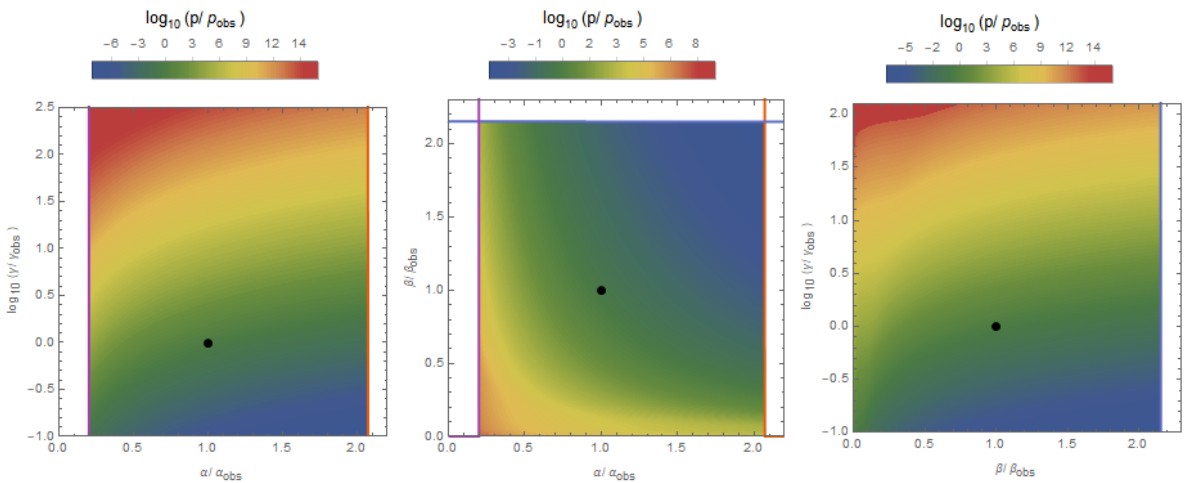

**Figure 6.** Distribution of observers from imposing the tidal locking condition.

### 4.5. Are Convective Stars Habitable?

In our universe, stars are divided into two broad classes, in accordance with the dominant mode of energy transfer between layers [76]. Stars below a certain size $0.35 M_\odot$ are convective, in that stellar material physically moves in order to attain local thermal equilibrium. Stars above $1.5 M_\odot$ are radiative, in that heat is transfered with relatively little radial mixing. Evidently, our sun lies between these two regimes, and accordingly it transfers heat using both methods. These two types of stars have a markedly different behavior, which may affect the overall habitability of the system.

Given the paucity of direct evidence, it is currently unknown how the heat transport of a star affects its habitability. Several reasons have been given to suspect that convective stars are indeed uninhabitable, all stemming from their pronounced churning. This induces strong XUV flux [77], flares [78] and space weather [79], all of which are capable of contributing to a severe level of atmospheric erosion of a planet orbiting within the habitable zone. As a counterpoint, these flares have been suggested to result in periods of potentially increased productivity in [80]. While there is a considerable effort currently devoted to determining whether these convective processes preclude life [81,82], here we may examine the consequences of adopting the viewpoint that convective stars are uninhabitable, and determine whether this is consistent with the multiverse framework.

---

[4]  The reader may object that even if a star's planets become tidally locked before it expires, they may remain tidally unlocked for sufficient time for complex life to develop. We have adopted this more rough criterion to make this section self contained, but one may instead compare it to the biological time discussed in the previous subsection, for example. If this is done, they would find $\lambda_{\text{TL}} = 3556 \alpha^{47/17} \beta^{12/17} \gamma^{-4/17}$, which does not alter the conclusions by much.

We take the threshold for stellar convection from [57], based off the dominance of Thomson scattering, and updated to incorporate our radial and luminosity dependence:

$$\lambda_{\text{conv}} = 0.57\,\alpha^3\,\beta\,\gamma^{-1/2} \tag{20}$$

Note the resemblance of this quantity to that of the photosynthesis condition from Equation (7). This striking fact traces its origins back to the original interpretation of this condition in [56], where it was assumed that the existence of both types of star is essential to life. The apparent coincidence of these two conditions is not as mysterious as it may first appear, either: Convection occurs in stars when their surface temperature drops below the point where molecular bonds are able to form, and so stars that exhibit marginal convection will have their surface temperatures automatically set by molecular energies. Then, this dichotomy between the behavior of small and large stars is a generic feature in any universe where photosynthesis is possible.

Using this condition gives:

$$\mathbb{H}_{\text{conv}} = N_\star\,f_{\text{conv}} \propto \alpha^{-3/2}\,\beta^{3/4}\,\gamma^3 \min\left\{1, .285\,\alpha^{-2.03}\,\beta^{-2.36}\,\gamma^{0.68}\right\} \tag{21}$$

The parameter dependence is very similar to the tidal locking case, with preference for small values of $\alpha$ and $\beta$ tamed by eventually entering a regime of parameter space where no stars are purely convective. Due to the strong similarities between these two scenarios, we refrain from plotting the probability distributions for this criterion. This gives probabilities of our observed values as:

$$\mathbb{P}(\alpha_{\text{obs}}) = 0.16, \quad \mathbb{P}(\beta_{\text{obs}}) = 0.41, \quad \mathbb{P}(\gamma_{\text{obs}}) = 2.6 \times 10^{-7}, \tag{22}$$

From here we see that this criterion performs about the same, though overall slightly better, than the tidal locking criterion.

### 4.6. Is Habitability Dependent on Entropy Production?

Up until now, the various hypotheses we have considered have failed to account for the observed values of our physical constants. This indicates that treating all stars that meet some threshold criteria as equally habitable may be the wrong approach. Though extensions to this simplistic scheme fall under the purview of the other factors in the Drake equation, which will be dealt with in subsequent papers, we take this opportunity to report on a prescription which manages to bring all predicted values into accord with observation.

The successful habitability criteria is that the presence of life should be proportional to the total entropy processed by a planet over its lifetime. On Earth, this is predominantly given by the downconversion of sunlight to lower frequencies, which in the process generates biologically useful chemical energy. The amount available will then depend on the total number of photons generated by the host star over its lifetime, as well as the solid angle subtended by the planet that collects them. For this, we specify to terrestrial mass planets that orbit within the temperate zone, as outlined in the Appendix A.

The reason one might consider this to play an important factor is that entropy production play a key role in regulating biosphere size [83,84], as evidenced by the fact that Earth's biosphere operates close to the theoretical limit for how much information can be processed. There are a number of subtleties in this argument that we do not address here, but will be dealt with in full in [22]. It suffices for the present purposes to merely introduce this criterion and demonstrate that it yields the desired results. One point we do wish to make, however, is that it seems inextricably linked with the necessity of photosynthesis: If the star only produced photons that could not be used for chemical energy, they would all be instantly recycled as waste heat, rather than contributing to the biosphere. Therefore, we do not consider this criterion in isolation ever, but only in conjunction with the photosynthesis requirement.

To this end, we estimate the total amount of entropy incident on a planet situated a temperate distance from its host star as:

$$\Delta S_{\text{tot}}(\lambda) \sim \frac{L_\star}{T_\star} \frac{R_{\text{terr}}^2}{4\,a_{\text{hab}}^2} t_\star \sim \frac{\alpha^{17/2}\,\beta^2}{\lambda^{119/40}\,\gamma^{17/4}} \sim 10^{54} \tag{23}$$

In this case, habitability will not simply be proportional to the fraction of stars meeting some certain criteria, but instead each star must be weighted by its entropy production. This will yield:

$$\mathbb{H}_S \propto \frac{\alpha^{203/80}\,\beta^{797/160}}{\gamma^{5/4}} \left( \min\left\{1, 0.45\frac{L_{\text{fizzle}}}{1100\text{ nm}} Y^{1/4}\right\}^{9.11} - \min\left\{1, 0.16\frac{L_{\text{fry}}}{400\text{ nm}} Y^{1/4}\right\}^{9.11} \right) \tag{24}$$

With $Y$ defined as before in Equation (10). This greatly ameliorates the smallness of $\gamma$ by virtue of the prefactor being very close to a scale invariant distribution. The distribution of observers is displayed in Figure 7.

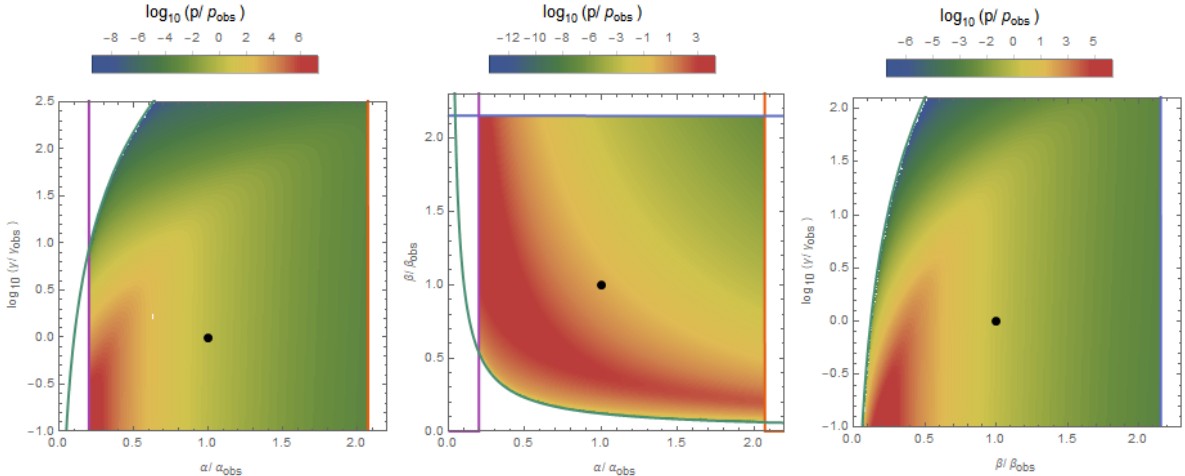

**Figure 7.** Distribution of observers from imposing the entropy condition.

Using the optimistic estimates for the photosynthetic range, the corresponding probabilities are:

$$\mathbb{P}(\alpha_{obs}) = 0.19, \quad \mathbb{P}(\beta_{obs}) = 0.45, \quad \mathbb{P}(\gamma_{obs}) = 0.32 \tag{25}$$

This is the first fully satisfactory synthesis of habitability criteria that is consistent with the multiverse hypothesis. It has implications for the distribution of observers that may be eventually tested: We should expect to find complex life in those locales with the most amount of entropy production. While fully determining the places this distinguishes will rely on an in-depth analysis, this would include planets that orbit more active stars, for longer, and able to collect more incident radiation.

## 5. Discussion: Comparing 40 Hypotheses

Until this point, we have considered the number of observers throughout universes with different microphysical constants and, weighing against the expected relative frequencies of such universes in a generic multiverse context, have determined the probability of measuring the three values of our constants as they are. Our findings show that these probabilities depend sensitively on the precise requirements for habitability that are assumed, as we have demonstrated by separately considering the expectations that complex life is proportional to the number of stars, that it is dependent on photosynthesis, the absence of tidal locking, that it can only arise around tame stars, that it requires a certain length of time to develop, and that its presence is proportional to the total amount of entropy processed by the system.

It is worth pausing to reflect on why this sensitivity should occur: Our estimates show that the number of observers in a universe is much more sensitive to the parameters than the underlying distributions we have taken. Due to this, our location in the multiverse will be dictated more by the actual requirements of life, rather than the availability of universes with those particular features. This may be contrasted with some of the cosmological parameters, where the underlying distribution can be exponentially sensitive, so that the expectation is simply to be in the most abundant locale, rather regardless of its habitability (recall that the volume fraction of our universe that is conventionally habitable is perhaps $10^{-40}$). More poetically, for the microphysical parameters we expect to be in 'the best of all possible worlds', whereas for the cosmological parameters, we expect to be in 'the cheapest of all possible worlds'.

However, being independent criteria, these may all also be considered in conjunction, yielding at this stage 40 distinct potential criteria for habitability. Because the effect of each condition was to restrict the range of habitable stellar masses, when taken together some effects will be more dominant that others in certain regions of parameter space. The various stellar thresholds are displayed in Figure 8.

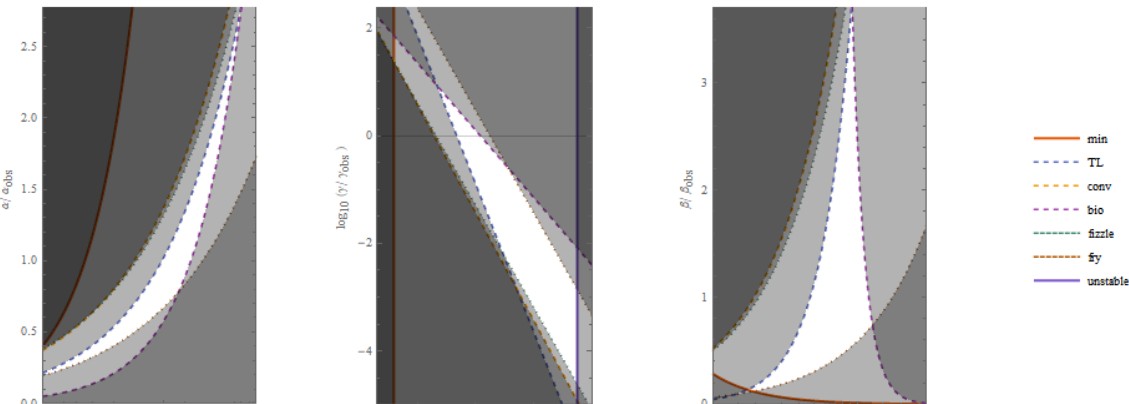

**Figure 8.** Habitable range of stellar masses for various choices of requirements. The shaded regions are treated as inhospitable for the various habitability assumptions.

We also take this opportunity to use a more realistic initial mass function, since including multiple criteria quickly make it extremely difficult to present results in an analytic form anyway. This takes into account the feedback that stars have on the collapsing protostellar dust cloud, which serves as a regulating mechanism guaranteeing that stars which are formed are within an order of magnitude or two of the scale $M_0 = (8\pi)^{3/2} M_{pl}^3 / m_p^2$. The resultant initial mass function from this process resembles a broken power law with a transition scale of 0.2 $M_\odot$ set by the minimum stellar mass; an analytic expression for this scale was presented in [85]. Various parameterizations of this distribution appear in the literature, but here we use the one of [86], which can be simply implemented as:

$$p_{\text{Masch}}(\lambda) = p_{\text{IMF}}(\lambda) \frac{8.94}{\left(1 + \left(\frac{5.06\lambda_{\min}}{\lambda}\right)^{\beta_{\text{IMF}}-1}\right)^{1.4}} \tag{26}$$

Employing this usually alters the probabilities we find by $\mathcal{O}(10\%)$ and occasionally by a factor of $\mathcal{O}(2)$, which is not enough to change any of our conclusions qualitatively.

In Table 2 we enumerate the full list of probabilities without including the entropy condition. The most salient feature of this list is that the probability of measuring our strength of gravity is never more that 0.0114 ($2.5\sigma$), universes with larger values being by our considerations more favorable than ours. We remind the reader that at this stage, we have restricted our attention to only the first factor of

the Drake equation, and so we may not find it surprising that we have not captured enough detail to yield a coherent account of our observations.

Other features may be noted if one examines the chart for long enough. There is a high degree of 'epistasis', to borrow a term from genetics: The inclusion of several requirements often does not influence the final result in a naively multiplicative manner. This can be seen by the inclusion of the biological timescale and tidal locking conditions, for instance: In isolation neither affect $\mathbb{P}(\alpha_{\mathrm{obs}})$ to a strong degree, whereas in conjunction $\mathbb{P}(\alpha_{\mathrm{obs}})$ is decreased by an order of magnitude. Including any additional habitability factors in isolation only decreases $\mathbb{P}(\beta_{\mathrm{obs}})$, and yet when combined the effects are not nearly as pronounced. However, we caution against interpreting from this observation, since none of these criteria are fully satisfactory.

**Table 2.** Probabilities of observing our values of parameters for various habitability hypotheses. Here the shorthands are photo: Photosynthesis criterion, TL: Tidal locking, conv: Convective stars, and bio: The biological timescale criterion.

| Criteria | $\mathbb{P}(\alpha_{\mathbf{obs}})$ | $\mathbb{P}(\beta_{\mathbf{obs}})$ | $\mathbb{P}(\gamma_{\mathbf{obs}})$ |
|---|---|---|---|
| number of stars | 0.198 | 0.437 | $4.15 \times 10^{-7}$ |
| bio | 0.281 | 0.116 | $2.52 \times 10^{-5}$ |
| conv | 0.183 | 0.426 | $3.1 \times 10^{-7}$ |
| conv bio | 0.0564 | 0.159 | $2.59 \times 10^{-5}$ |
| TL | 0.152 | 0.37 | $2.34 \times 10^{-7}$ |
| TL bio | 0.0101 | 0.413 | $8.27 \times 10^{-5}$ |
| TL conv | 0.152 | 0.37 | $2.34 \times 10^{-7}$ |
| TL conv bio | 0.0101 | 0.413 | $8.27 \times 10^{-5}$ |
| photo | 0.439 | 0.183 | $8.16 \times 10^{-7}$ |
| photo bio | 0.0631 | 0.103 | $1.7 \times 10^{-5}$ |
| photo conv | 0.439 | 0.183 | $8.06 \times 10^{-7}$ |
| photo conv bio | 0.0637 | 0.104 | $1.7 \times 10^{-5}$ |
| photo TL | 0.48 | 0.232 | $6.88 \times 10^{-7}$ |
| photo TL bio | 0.0352 | 0.281 | 0.000139 |
| photo TL conv | 0.48 | 0.232 | $6.88 \times 10^{-7}$ |
| photo TL conv bio | 0.0352 | 0.281 | 0.000139 |
| yellow | 0.486 | 0.162 | $8.78 \times 10^{-7}$ |
| yellow bio | 0.0351 | 0.102 | $1.72 \times 10^{-5}$ |
| yellow conv | 0.486 | 0.162 | $8.78 \times 10^{-7}$ |
| yellow conv bio | 0.0351 | 0.102 | $1.72 \times 10^{-5}$ |
| yellow TL | 0.0303 | 0.0308 | $1.63 \times 10^{-6}$ |
| yellow TL bio | 0.324 | 0.335 | 0.0114 |
| yellow TL conv | 0.0303 | 0.0308 | $1.63 \times 10^{-6}$ |
| yellow TL conv bio | 0.324 | 0.335 | 0.0114 |

Additionally of note is that the inclusion of the photosynthesis condition renders the convective star condition almost completely superfluous, which makes sense in light of the fact that convective stars are always slightly lighter than the minimal photosynthetic star. This addresses an otherwise puzzling feature of our universe that arises if one believes convective stars are uninhabitable, which is why so many stars have this property, seemingly wasting the majority of opportunities for life to develop. If one simultaneously takes the viewpoint that photosynthesis is necessary for complex life, however, this puzzle is resolved because convective stars are a generic byproduct of the fact that photosynthetic light is just barely able to break chemical bonds.

Most importantly, however, is the observation that the criteria used change the probabilities by several orders of magnitude. The biological timescale condition, in particular, increases $\mathbb{P}(\gamma_{\mathrm{obs}})$ by up to a factor of 353 by limiting the range of the strength of gravity. The spread in the probabilities of our observed values are 48, 14, and 49,000 for $\alpha$, $\beta$ and $\gamma$, respectively. The overall spread of the product of these values is 813,000. This gives the indication that the relative confidence that can be gained about certain habitability conditions will be of the same order of magnitude.

Table 3 incorporates the entropy condition as well. As can be seen, this brings all probabilities well into agreement with observations, irrespective of the inclusion of the various other hypotheses.

Interestingly, the spread of values once the entropy condition is included is much narrower than without, being 3.8, 1.3 and 1.5 for $\alpha$, $\beta$, and $\gamma$ respectively. The spread of the product of these values is now 3.6. This tempering is due to the fact that the entropy and photosynthesis conditions place such restrictive bounds on habitability, the effects of the other conditions (which primarily effect other regions of parameter space) play little role. As of now, this may seem somewhat disappointing, as the multiverse hypothesis has seemingly nothing to say about the role of tidal locking, stellar lifetime, convective habitability, or range of photosynthetic wavelengths, as long as one imposes the photosynthesis and entropy conditions as necessary for complex life. Indeed, one should not expect to come away with strong expectations for every proposed requirement based off these arguments. However, more information can in fact be gleaned about which hypotheses are viable based off a few additional considerations regarding our actual location within our own universe: This will be explored fully in [22].

**Table 3.** Probabilities of observing our values of parameters for various habitability hypotheses with entropy condition (denoted by S) included. The other shorthands are the same as above.

| Criteria | $\mathbb{P}(\alpha_{\mathbf{obs}})$ | $\mathbb{P}(\beta_{\mathbf{obs}})$ | $\mathbb{P}(\gamma_{\mathbf{obs}})$ |
|---|---|---|---|
| photo S | 0.24 | 0.386 | 0.376 |
| photo bio S | 0.178 | 0.414 | 0.426 |
| photo conv S | 0.256 | 0.401 | 0.368 |
| photo conv bio S | 0.191 | 0.433 | 0.421 |
| photo TL S | 0.394 | 0.446 | 0.356 |
| photo TL bio S | 0.278 | 0.465 | 0.453 |
| photo TL conv S | 0.394 | 0.446 | 0.356 |
| photo TL conv bio S | 0.278 | 0.465 | 0.453 |
| yellow S | 0.191 | 0.45 | 0.317 |
| yellow bio S | 0.125 | 0.486 | 0.38 |
| yellow conv S | 0.191 | 0.45 | 0.317 |
| yellow conv bio S | 0.125 | 0.486 | 0.38 |
| yellow TL S | 0.481 | 0.396 | 0.44 |
| yellow TL bio S | 0.343 | 0.476 | 0.31 |
| yellow TL conv S | 0.481 | 0.396 | 0.44 |
| yellow TL conv bio S | 0.343 | 0.476 | 0.31 |

Remember that, of the 40 possible conditions we started with, less than half have turned out to be compatible with the multiverse hypothesis. Considering additional habitability criteria will yield similarly strong predictions for multiple other leading schools of thought about what conditions are necessary for complex life. This demonstrates the power of this method of reasoning: We have utilized this method to uncover readily discoverable facts about the world that are currently unknown. The true conditions for habitability will eventually be found, and sooner than one might be prepared for: Upcoming experiments probing the solar system and galaxy promise to shed light on these issues, and inform our understanding of life's place in the universe, and, depending on their findings, multiverse.

**Funding:** This research received no external funding.

**Acknowledgments:** I would like to thank Fred Adams and Alex Vilenkin for useful discussions.

**Conflicts of Interest:** The author declares no conflict of interest.

## Appendix A. Stellar Properties

Throughout, we have made use of how the main features of stars scale with mass. These are well known, as can be found in [23,42], for example, who take a particular emphasis on dependence on physical constants. We restrict our summary to main sequence stars.

One of the most important stellar characteristics is its luminosity, which is given by:

$$L_\star = 9.7 \times 10^{-4} \lambda^{q_L} \frac{m_e^2 \, M_{pl}}{\alpha^2 \, m_p} \tag{A1}$$

The dependence on mass is sometimes given as a broken power law, which reflects the fact that the opacity inside the star is set by different scattering processes depending on the temperature and pressure. Here, we neglect this subtlety, as it does not greatly affect our analysis other than making it far less amenable to analytic study, and take the value $q_L = 3.5$ from [76]. This most accurately characterizes smaller stars, which dominate the population, and so are most important to consider.

Equally important is the lifetime of a star, which can be found through the approximate scaling relation $t_\star \approx \epsilon_{\mathrm{nuc}} M_\star / L_\star$, where $\epsilon_{\mathrm{nuc}}$ is the energy yield per nucleon:

$$t_\star = 85.6 \frac{\alpha^2}{\lambda^{5/2}} \frac{M_{pl}^2}{m_p \, m_e^2} \tag{A2}$$

where we can see the characteristic scaling that massive stars live for a shorter duration than less massive stars.

The radius of a star is observed to scale as:

$$R_\star = 108.6 \, \lambda^{q_\xi} \frac{M_{pl}}{\alpha^2 \, m_p} \tag{A3}$$

Here, $q_{\bar\xi} = 4/5$ [87]. Using this result, we can derive the star's surface temperature to be:

$$T_\star = 0.014 \, \lambda^{\frac{q_L - 2q_\xi}{4}} \frac{\alpha^{1/2} \, m_e^{1/2} \, m_p^{3/4}}{M_{pl}^{1/4}} \tag{A4}$$

For this, the expression for interior temperature of a star given in [23] was used. This estimates the interior temperature of the star by imposing the condition $\sqrt{T/m_p} \sim \alpha$ in order for thermal effects to balance out the energetic suppression that comes from the Coulomb barrier in reactions. As can be seen, the dependence of temperature on stellar mass is actually quite weak, $T \propto \lambda^{19/40}$, signifying that all stars emit light in approximately the same wavelength regime. This lends credence to the claim that a star's suitability for photosynthesis is largely independent of mass, but rather only contingent on the relation imposed on physical constants.

The stellar temperature can be compared to the temperature needed for life: This is commonly defined as the value for which liquid water is possible, but more generically, it can be identified as the temperature that matches the energy levels of typical molecular bonds, so that life is free to manipulate and store energy by subtly rearranging local chemical conditions. This requirement was identified in [23] to give:

$$T_{\mathrm{mol}} = 0.037 \frac{\alpha^2 \, m_e^{3/2}}{m_p^{1/2}} \tag{A5}$$

This is a factor $(m_e/m_p)^{1/2}$ smaller than the Rydberg temperature that governs atomic ionization due to the lower energy vibrational modes of the molecules. Additionally, we include the factor $\epsilon_T \sim 0.037$, which encodes "the abhorrent details of chemistry that are omitted" from [23].

This defines the typical timescale for molecular processes as $t_{mol} = 1/T_{mol}$. This definition is based off the rotational frequencies of molecules—it can be viewed as the time it takes energy to redistribute throughout a molecule. This can be compared with the timescale set by the mean free path over the sound speed, $L_{mfp}/c_s$, the typical time between molecular interactions. For room temperature solutions, these scales differ only by a factor of $\beta^{-1/2}$, and so the distinction will be unimportant in this setting, but crucial for other purposes. These can also be compared with the atomic timescale used in [88] to set an upper bound on $\gamma$, which nevertheless produces similar results.

The traditional habitable orbital distance from the star can be found by demanding that the planet be at the habitable temperature. Though this depends on many planetary factors, these will be dealt with more properly in [29]. For now we content ourselves with a simple blackbody estimate, which yields $a = (T_\star/T)^2 R_\star/2$. Then:

$$a_{temp} = 7.6\,\lambda^{q_L/2}\,\frac{m_p^{1/2}\,M_{pl}^{1/2}}{\alpha^5\,m_e^2} \tag{A6}$$

Finally, the radius of a terrestrial planet can be found by the condition that the escape velocity is of the same order as the thermal velocity for the molecular temperature, which yields:

$$R_{terr} = 3.6\,\frac{M_{pl}}{\alpha^{1/2}\,m_e^{3/4}\,m_p^{5/4}} \tag{A7}$$

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
