# Peer review of "Multiverse Predictions for Habitability: The Number of Stars and Their Properties"

_universe, doi:10.3390/universe5060149_

Reviewer 1 Report

The article "Multiverse predictions for habitability: the number of stars and their properties" presents an original approach to predict habitability within the multiverse context. The author studies the implications of the multiverse hypothesis (MH) to 40 habitability conditions, showing that only the minority of them are compatible with the MH.

One interesting aspect of this treatment is the fact that variables not directly used are integrated over, thus removing the common problem in fine-tuning studies related to the degeneracy between parameters. I consider this one of the major strengths of the article.

I believe this article is within the scope of this journal and it is of interest to a very broad audience. I find the article relatively well-written, though it is not always very pedagogical. Therefore, my first suggestion would be for the author to take a closer look at the manuscript and rewrite some sentences that are too long (leading to ambiguities), or too complex.

There are a few points that should be addressed before it is suitable for publication. I list them below.

1.  Lines 7-8: The author says that the multiverse can be elevated to a predictive scientific framework.
First, I believe this statement is too strong.
Second, I fail to see how observations in our universe can make real scientific predictions about other universes. This link is far from obvious and sounds to me a bit misleading.
Moreover, to say "predictive scientific framework" does not seem appropriate (from a philosophical standpoint) considering the analyses that has been done, and the reasoning that has been presented. I fail to see the predictive power of the method presented without having a very large sample of forms of life, a precise definition for life. Even if we did have it, I seriously doubt these predictions can be safely extrapolated to the multiverse. Therefore, I would suggest the author to weaken this statement.

2. Lines 65-68 (and Fig.2):
The statements "if life needs somethings, we expect our universe to be good at it" and "if our universe is good at something, we expect that to be important for life"
While the former seems indisputable, the latter is not necessarily true. I understand this is a fundamental assumption of the whole analysis, as emphasised in Fig. 2, but it is not obvious. I think it is important to stress that this is an assumption rather than a logical conclusions, as the sentence suggests.

3. Line 91: I do not conclude from the text that there is "strong evidence against the MH".

4. Lines 91-92: If the multiverse is correct, we would not necessarily expect ALL habitability criteria to be met. In fact, since we don't even know what life is, this statement is inaccurate.

5. Lines 132-134: Can the strength of gravity really be changed that much? There is an interplay between the cosmological constant (lambda) and G. This determines the timescale for structure formation. Lambda depends on G, but also on the the vacuum density. Therefore, the timescales involved in producing galaxies and stars in this alternative scenario should be compared with the time it takes for life to emerge. I am not convinced that that one can play that much with the strength of gravity.
Similar arguments have been provided by Carr & Rees, Nature 278 (1979) 605. They use the gravitational fine-structure constant, alpha_G.

6. Lines 206-217: As it is well known, the choice of priors is extremely important. Although the  author claims that the choice of priors is suitable, different (equally suitable) priors may lead to completely different results. It would be instructive to compare the current results with results obtained using a different prior (maybe only for one or two cases).

7. Lines 277-280: This argument has been discussed in Loeb, Batista, Sloan JCAP 08 (2016) 040.

8. In lines 344-356 the `photosynthesis criterion' is introduced.
From an evolutionary perspective, it is just natural that life will evolve in such a way as to match the wavelengths of the host star. The choice of 750 nm is completely understandable if one consider the case of water. However, if the yellow criterion is defined considering the 750 nm wavelength, it excludes forms of life that might be living elsewhere -- and we have many examples here on Earth -- like hydrothermal vents, underwater creatures, etc.
Most importantly, I am concerned about the importance of photosynthesis for life. While the role it plays in Earth's biosphere is undeniable, it may be a bit too restrictive to require photosynthesis for life to emerge. There may be other mechanisms to regulate the habitability of a planet for forms of life similar to those known to us, that does not necessarily require photosynthesis.
My suggestion is to improve the justification for using photosynthesis as a necessary condition for habitability.

9. Line 379: In the subsection "Is photosynthesis possible around red dwarfs?", part of the presented discussion is presented in more detail in Shields, Ballard, Johnson Phys. Rep. 663 (2016) 1.

10. Line 460: Attributing the scattering to Thompson, instead of Thomson, is a rather common and widespread historical error. It should read "Thomson scattering", referring to J. J. Thomson's 1906 work.

12. Line 489-491: To say that "it would do no good for a star to produce abundant amounts of photons if they could not be put to use" is a teleological argument. I strongly advise against such arguments as one cannot easily stand by them.
The star produces photons. The fate of these photons is independent of the star. Naturally, there are optimal cases in which they are recycled. But this is "advantageous" only to whatever process is using them, not to the star.

13. Lines 561-568: The author says that less than half of the conditions are met by the MH. This, however, strongly depends on the values of P(alpha,beta,gamma). Is there a way to set bounds to P (something like error bars), using what we know about life forms, the most resilient ones, etc?
I am not suggesting this to be done immediately. I would simply like to know if the fraction quotes (about half) changes significantly by using wider distributions and possibly other priors.

14. As a general remark, high-resolution figures would be better.

In summary, I find the work original and satisfactorily written. At the moment there are some imprecisions, some overly optimistic arguments, and some points that require clarifications. However, after the points I raised are addressed, I believe it will be suitable for publication.

Author Response

I thank the reviewer for their suggestions on the draft, and find that incorporating them does make the paper better.

I believe this article is within the scope of this journal and it is of interest to a very broad audience. I find the article relatively well-written, though it is not always very pedagogical. Therefore, my first suggestion would be for the author to take a closer look at the manuscript and rewrite some sentences that are too long (leading to ambiguities), or too complex.

Noted.  I checked all sentences over 2.5 lines in length, and amended any that seemed too complex.

There are a few points that should be addressed before it is suitable for publication. I list them below.

1.  Lines 7-8: The author says that the multiverse can be elevated to a predictive scientific framework.
First, I believe this statement is too strong. 
Second, I fail to see how observations in our universe can make real scientific predictions about other universes. This link is far from obvious and sounds to me a bit misleading. 
Moreover, to say "predictive scientific framework" does not seem appropriate (from a philosophical standpoint) considering the analyses that has been done, and the reasoning that has been presented. I fail to see the predictive power of the method presented without having a very large sample of forms of life, a precise definition for life. Even if we did have it, I seriously doubt these predictions can be safely extrapolated to the multiverse. Therefore, I would suggest the author to weaken this statement. 

I do feel that the abstract was worded a bit too strongly, and it has been amended.

However, I want to push back on the reviewer’s skepticism a bit.  When I say that the multiverse can be made predictive, what I mean is that the multiverse is only a coherent framework if it explains our presence in this universe.  Since this depends on the assumptions for what life needs, we can select out preferred habitability criteria which are compatible with the multiverse.  These can eventually be tested.  Yes, it will take a very large sample of lifeforms, but consider that LUVOIR is slated to measure the atmospheres of 100 Earthlike planets orbiting main sequence stars in the temperate zone, and a factor of 10 more for planets in general.  If life is even somewhat common, we will get a large sample within a few decades.  If it’s rarer, it may take 100 years or longer.  Either way, these predictions will eventually be testable.

True, I do not have a prescriptive definition of life, and nobody may ever have one, but my hope is that this will not be strictly necessary to make progress.  It may be required to address the thorniest questions, but for more modest questions, such as whether life can arise on tidally locked planets, a general definition is not required.

2. Lines 65-68 (and Fig.2): 
The statements "if life needs somethings, we expect our universe to be good at it" and "if our universe is good at something, we expect that to be important for life"
While the former seems indisputable, the latter is not necessarily true. I understand this is a fundamental assumption of the whole analysis, as emphasised in Fig. 2, but it is not obvious. I think it is important to stress that this is an assumption rather than a logical conclusions, as the sentence suggests.

The confusion here is my fault, and comes from the desire to make the logic of this paper as slogan-esque as possible in order for it to be accessible to a general audience.  By “our universe is good at H” I meant two things: that by taking H, our presence here is probable, and by taking not H, our presence here is improbable.  I had not quite elucidated this before, so this has been amended in the updated version.

3. Line 91: I do not conclude from the text that there is "strong evidence against the MH".

I would like to try to convince the reviewer on this point: There is an analog argument in Loeb JCAP 05 (2006) 009 for the cosmological constant.  Current anthropic bounds conditioned on the existence of large galaxies allow for Lambda to be 1-2 orders of magnitude larger.  If only dwarf galaxies are necessary for life, it could be larger still, making our presence in this universe highly improbable.  Therefore, if we find that life can indeed arise in dwarf galaxies, the multiverse hypothesis will be ruled out to a high degree of confidence.  I am implementing the same logic here, only on a planetary rather than a galactic scale, as the observational prospects are much better.

4. Lines 91-92: If the multiverse is correct, we would not necessarily expect ALL habitability criteria to be met. In fact, since we don't even know what life is, this statement is inaccurate. 

This statement may have caused confusion, so I amended it in the text.  I only meant that if I claim that both A and B are true, I am only right if both are true.  If either one is wrong or if they are both wrong, then I was wrong.

5. Lines 132-134: Can the strength of gravity really be changed that much? There is an interplay between the cosmological constant (lambda) and G. This determines the timescale for structure formation. Lambda depends on G, but also on the the vacuum density. Therefore, the timescales involved in producing galaxies and stars in this alternative scenario should be compared with the time it takes for life to emerge. I am not convinced that that one can play that much with the strength of gravity.
Similar arguments have been provided by Carr & Rees, Nature 278 (1979) 605. They use the gravitational fine-structure constant, alpha_G.

Here, I have treated Lambda as an independent variable, so that it may in effect be held fixed in order to provide a fixed H_0 / m_p.  This was done purely for convenience, so I could focus on the 3 quantities I consider in the article.  I agree that it is important to incorporate this additional variable, as it may have a strong influence on the probabilities of our observations.  Indeed, I have plans to write a future paper on galactic habitability with my formalism that does just this.  However, for the moment this will have to be left to future work.  If attention is restricted to stellar habitability, then the bound I quote remains true.  I did look very hard through the literature to find a stronger bound, but have not found any so far.

6. Lines 206-217: As it is well known, the choice of priors is extremely important. Although the  author claims that the choice of priors is suitable, different (equally suitable) priors may lead to completely different results. It would be instructive to compare the current results with results obtained using a different prior (maybe only for one or two cases).

A uniform prior is now investigated for the first habitability criterion, and a footnote is added on line 300.

7. Lines 277-280: This argument has been discussed in Loeb, Batista, Sloan JCAP 08 (2016) 040.

Reference added.

8. In lines 344-356 the `photosynthesis criterion' is introduced. 
From an evolutionary perspective, it is just natural that life will evolve in such a way as to match the wavelengths of the host star. The choice of 750 nm is completely understandable if one consider the case of water. However, if the yellow criterion is defined considering the 750 nm wavelength, it excludes forms of life that might be living elsewhere -- and we have many examples here on Earth -- like hydrothermal vents, underwater creatures, etc.
Most importantly, I am concerned about the importance of photosynthesis for life. While the role it plays in Earth's biosphere is undeniable, it may be a bit too restrictive to require photosynthesis for life to emerge. There may be other mechanisms to regulate the habitability of a planet for forms of life similar to those known to us, that does not necessarily require photosynthesis. 
My suggestion is to improve the justification for using photosynthesis as a necessary condition for habitability. 
I remind the reviewer that the photosynthesis criterion here is treated as a hypothesis: some people in the literature have claimed that photosynthesis in the narrow range around 750nm is essential for complex life, others claim the range should be broadened, and others still claim it is irrelevant.  The point of my paper is not to commit to any one dogma, but to incorporate each separately, so see which, if any, are compatible with the multiverse hypothesis.  As such, each of the different photosynthesis criteria are only included in 1/3 of the list of habitability criteria I consider.

I feel I have given plenty of justification for treating photosynthesis as necessary in lines 319-336, where I include the reasons that it allows 1000 times more energy to be harvested than chemosynthesis, that it is the source of essentially all biotic matter on Earth today, that an oxygenated atmosphere is necessary for animal life, and the colonization of land.  In fact, I know of no further justifications for why photosynthesis is necessary.  Having said that, I sympathize with the view that it may not be needed, and think this viewpoint deserves equal attention.

9. Line 379: In the subsection "Is photosynthesis possible around red dwarfs?", part of the presented discussion is presented in more detail in Shields, Ballard, Johnson Phys. Rep. 663 (2016) 1.

Reference added.

10. Line 460: Attributing the scattering to Thompson, instead of Thomson, is a rather common and widespread historical error. It should read "Thomson scattering", referring to J. J. Thomson's 1906 work.
Fixed.

12. Line 489-491: To say that "it would do no good for a star to produce abundant amounts of photons if they could not be put to use" is a teleological argument. I strongly advise against such arguments as one cannot easily stand by them.
The star produces photons. The fate of these photons is independent of the star. Naturally, there are optimal cases in which they are recycled. But this is "advantageous" only to whatever process is using them, not to the star. 

Reworded to remove all traces of teleology from the statement.

13. Lines 561-568: The author says that less than half of the conditions are met by the MH. This, however, strongly depends on the values of P(alpha,beta,gamma). Is there a way to set bounds to P (something like error bars), using what we know about life forms, the most resilient ones, etc?
I am not suggesting this to be done immediately. I would simply like to know if the fraction quotes (about half) changes significantly by using wider distributions and possibly other priors. 

This is an interesting point.  The second and third papers in this series deal more with the biological aspects of habitability, such as the temperature tolerance of organisms, but it is not phrased in the terms the reviewer mentions.  This will definitely be something to consider for future work.

14. As a general remark, high-resolution figures would be better. 

Unfortunately, higher quality versions of these plots looked terrible in almost every pdf viewer except chrome, actually took time to load each page, and made the file size enormous.  These were the best I could come up with that are a little rasterized but still look good.

In summary, I find the work original and satisfactorily written. At the moment there are some imprecisions, some overly optimistic arguments, and some points that require clarifications. However, after the points I raised are addressed, I believe it will be suitable for publication.

Reviewer 2 Report

This is a clearly written paper by an author who is plainly widely read in the literature of ‘anthropics’ and counterfactual physics. He presents (as summarized in Figure 1) an illuminating and distinctive way of addressing the ‘typicality’ of our universe, and the plausibility that it is the outcome o anthropic selection from an ensemble (a multiverse).

The paper is presented as one  of a series, which each address  one of the parameters in the Drake-type  equation  that determines the habitability of a universe.

Most of the quantities are presented (as is common in the literature of this subject since the pioneering work of Carter) in  terms of fundamental constants multiplied by some very uncertain number. But some data are very specific (eg those in Table 2 which depend on detaild properties of stellar atmosphere).

While the paper is readable and interesting, I wonder whether it is optimal to present separate papers discussing each parameter since they are clearly interlinked.

For instance nothing is said about nucleosynthesis. This requires, at the very least,  an early generation of massive stars, and constrains the stellar IMF in ways that would make its actual  form closer to optimal that it would be if the sole criterion was to maximize the number of stars.

As another instance of interconnection, the paper points out that the constraints on G are not at all severe. However, it would be straightforward to include the cosmological consequences of a different G : it can be shown that galactic masses would scale with G in the same way as the Chandrasekhar mass, so that the number of stars per galaxy is independent of G. (the H/He ratio is however sensitive to G). But the implications would depend on how the ‘size of the universe’ was constrained

Although the references are comprehensive, the author might help potential readers by including the very comprehensive recent review by Fred Adams (arXiv 1902. 03928)

Author Response

I thank the reviewer for their suggestions on the draft, and find that incorporating them does make the paper better.

Most of the quantities are presented (as is common in the literature of this subject since the pioneering work of Carter) in  terms of fundamental constants multiplied by some very uncertain number. But some data are very specific (eg those in Table 2 which depend on detaild properties of stellar atmosphere).

 It's true, the simple scaling arguments used do not do justice to the full intricacies of stellar physics, and so will only be useful for order of magnitude estimates.  It is worth pointing out that current stellar models disagree on how to implement the physics of diffusion, scattering, and convection at the 5% level (at least on how to encapsulate these properties in a 1d approximation), and so it may be premature now to go beyond the scaling arguments I employ.  I would be interested in going beyond these scaling arguments in the future, but the benefit of the approach I took was that it allowed me to explore many different habitability criteria quickly.  This allows to identify promising avenues (like this one) for targeted follow-up work.

While the paper is readable and interesting, I wonder whether it is optimal to present separate papers discussing each parameter since they are clearly interlinked.

It will interest the reviewer to know that the plan is to publish all four papers in this special edition of the journal.  The segmentation into separate papers was purely due to feasibility, as otherwise it would have been over 100 pages long.

For instance nothing is said about nucleosynthesis. This requires, at the very least,  an early generation of massive stars, and constrains the stellar IMF in ways that would make its actual  form closer to optimal that it would be if the sole criterion was to maximize the number of stars.

 On this point, this is considered in the second paper in this series.

As another instance of interconnection, the paper points out that the constraints on G are not at all severe. However, it would be straightforward to include the cosmological consequences of a different G : it can be shown that galactic masses would scale with G in the same way as the Chandrasekhar mass, so that the number of stars per galaxy is independent of G. (the H/He ratio is however sensitive to G). But the implications would depend on how the ‘size of the universe’ was constrained

Indeed, cosmological considerations can introduce additional upper bounds on G.  However, to be meaningful, the other cosmological parameters must be included in the analysis, which is not done here.  I plan on carrying this out in a future publication related to galactic habitability, but time and space considerations prevent me from incorporating it here.

Although the references are comprehensive, the author might help potential readers by including the very comprehensive recent review by Fred Adams (arXiv 1902. 03928)

Reference added.